# Triple-Negative Breast Cancer: Basic Biology and Immuno-Oncolytic Viruses

**DOI:** 10.3390/cancers15082393

**Published:** 2023-04-21

**Authors:** Michael L. Monaco, Omer A. Idris, Karim Essani

**Affiliations:** Laboratory of Virology, Department of Biological Sciences, Western Michigan University, Kalamazoo, MI 49008, USA; michael.l.monaco@wmich.edu (M.L.M.); omer.a.idris@wmich.edu (O.A.I.)

**Keywords:** triple-negative breast cancer, immuno-oncolytic viruses, cancer biology, virotherapy, combination therapy

## Abstract

**Simple Summary:**

Triple-negative breast cancer (TNBC) is an aggressive as well as the most dangerous form of breast cancer. Due to the lack of biomarkers that can be targeted by specific molecular therapeutics, treatment is usually limited to chemotherapy and surgery where applicable. Oncolytic viruses are biologicals that have been engineered in many cases to express different genes that can aid in the recruitment of immune cells to trigger antitumor immune responses in addition to the direct lysis of infected cancerous cells. In this review, the basic biology of TNBC is described, as are the many OVs and their different strategies to attempt to treat TNBC in vitro, in vivo, and in human trials.

**Abstract:**

Triple-negative breast cancer (TNBC) is the most lethal subtype of breast cancer. TNBC diagnoses account for approximately one-fifth of all breast cancer cases globally. The lack of receptors for estrogen, progesterone, and human epidermal growth factor 2 (HER-2, CD340) results in a lack of available molecular-based therapeutics. This increases the difficulty of treatment and leaves more traditional as well as toxic therapies as the only available standards of care in many cases. Recurrence is an additional serious problem, contributing substantially to its higher mortality rate as compared to other breast cancers. Tumor heterogeneity also poses a large obstacle to treatment approaches. No driver of tumor development has been identified for TNBC, and large variations in mutational burden between tumors have been described previously. Here, we describe the biology of six different subtypes of TNBC, based on differential gene expression. Subtype differences can have a large impact on metastatic potential and resistance to treatment. Emerging antibody-based therapeutics, such as immune checkpoint inhibitors, have available targets for small subsets of TNBC patients, leading to partial responses and relatively low overall efficacy. Immuno-oncolytic viruses (OVs) have recently become significant in the pursuit of effective treatments for TNBC. OVs generally share the ability to ignore the heterogeneous nature of TNBC cells and allow infection throughout a treated tumor. Recent genetic engineering has allowed for the enhancement of efficacy against certain tumor types while avoiding the most common side effects in non-cancerous tissues. In this review, TNBC is described in order to address the challenges it presents to potential treatments. The OVs currently described preclinically and in various stages of clinical trials are also summarized, as are their strategies to enhance therapeutic potential.

## 1. Introduction

### 1.1. Triple-Negative Breast Cancer (TNBC)

Breast cancer (BC) is the most diagnosed cancer in the United States for women, and is also the number two killer among cancers in women [1,2]. The biggest challenge with BC treatment lies in the heterogeneity of the tumors. There are three different receptors that define different BC subtypes: estrogen receptors (ERs, inside cells), progesterone receptors (PRs, inside cells), and human epidermal growth factor receptor 2 (HER-2, also named CD340, a cell surface receptor) [3,4]. Of all of the new cases of BC each year, approximately 10–20% are negative for all 3 receptors and are categorized as TNBC [5,6]; however, within TNBC itself, there are several different subtypes based on differential gene expression. These types are as follows: basal-like (BL1 and BL2), mesenchymal (M), mesenchymal-stem-cell-like (MSL), immunomodulatory (IM), and luminal androgen receptor (LAR) [7,8,9,10].

The BL1 and BL2 subtypes differ in that BL1 has elevated gene expression for DNA damage response, proliferation signaling, and cell cycle checkpoint loss, whereas BL2 has elevated growth factor (GF) signaling, glycolysis/gluconeogenesis signaling, and myoepithelial surface receptors. This suggests that cells of the BL1/2 subtypes may be of basal or myoepithelial origin. The M subtype has elevated expression of cell motility pathways, receptor interaction with the extracellular matrix (ECM), and cell differentiation pathways. The MSL subtype shares expression with the M subtype, but it also uniquely expresses pathways related to GF signaling (inositol phosphate metabolism, platelet-derived growth factor (PDGF), epidermal growth factor receptor (EGFR), calcium (Ca^2+^), G-protein-coupled receptor (GPCR), and extracellular-related kinase (ERK 1 and 2)), epithelial–mesenchymal transition (EMT), and Wnt/β-catenin. The MSL subtype also has a reduced expression of genes for cell proliferation. The IM subtype has an elevated expression of pathways involving immune cell signaling, cytokine signaling, antigen processing/presentation, and immune signal transduction. Finally, the LAR subtype has a heavily elevated expression of pathways related to hormone regulation, particularly with androgen receptor signaling (which can be approximately nine times as much as that of other TNBC subtypes) [8]. These extremely diverse genetic regulation conditions define tumor heterogeneity within TNBC, which helps to explain why TNBC is extremely difficult to treat in comparison with other BCs [8,11].

### 1.2. Development of Heterogeneity in TNBC

There have been a few theories to describe how heterogeneity in TNBC is established and maintained. The two main concepts are the cancer stem cell (CSC) hypothesis and the clonal evolution model (CEM). Other concepts, such as the deregulation of adult mammary stem cells (aMSCs), are described as contributing to developing TNBC in its initial stages [12,13,14,15].

The division of aMSCs gives rise to progenitors that further differentiate into basal or luminal progenitor cells. Under normal conditions, if basal or luminal progenitor cells are made, they further develop into mature basal or luminal cells; however, when the development of luminal progenitors is disrupted, cancer cells may develop. This is believed to occur via the improper regulation of Wnt/β-catenin pathways, transcription factors (TFs), such as Snail, and embryonic stem cell (ESC) markers (Sox2, Nanog, and Oct4). Combinations of genetic alterations to these pathways can make the progenitor cells de-differentiate and return to a proliferative state, which, when combined with other mutations, may result in TNBC [15].

The CSC model proposes that there is a hierarchy within cancers of tumorigenic cancer cells and their non-tumorigenic progeny. Within these subpopulations, the CSCs are hypothesized to be the driving forces behind the growth of a tumor, its progression, resistance to certain treatments, and metastasis [12,16,17,18,19]. This model does not address how to account for the massive variability in tumor heterogeneity between cancers of the same type in different individuals. Early characterizations of tumorigenic CSCs in BCs were via the expression of CD44 and not CD24 (including the opposite expression pattern); however, these characterizations have not held because many subtypes do not follow these marker patterns [12]. Moreover, the alteration of aldehyde dehydrogenase 1 (ALDH1) expression has also been independently reported to have tumorigenic potential in BCs [20].

The CEM presumes that there is one original clone that becomes cancerous, which then divides and, as carcinogenesis progresses, the cell as well as its subsequent progeny accumulate mutations in each cycle of clonal expansion, which results in the heterogeneity of the tumor. This model also assumes that there is no hierarchy among the cells and that, instead, natural selection determines which clones are successful and continue to divide. The cells that cannot undergo division are maintained as part of the tumor or will senesce, eventually being eliminated via necrosis [13,14]. Tumor heterogeneity may originate via some combination of the two models and several additional variables, including environmental factors, genetic background, and immune status. Hence, additional studies are needed to define the role of these contributing factors in individual cancers in order to formulate a more comprehensive model in the future.

### 1.3. TNBC Pathogenesis

Among the approximately 268,600 new cases of BC in the United States in 2019, about 10–20% of these were TNBC [21]. Estimates for 2022 suggested 287,850 new cases of BC in women of the United States, resulting in approximately 43,250 deaths [1]. TNBC, as a whole, is more aggressive, has more p53 mutations, has higher mitotic indices, has less defined nuclear pleomorphisms, has a higher average grade, and demonstrates higher rates of proliferation than typical BCs [22]. In addition to the above, TNBC has a high risk for relapsing in a patient following therapy, which factors into its high mortality compared to other BCs [23].

In the development and progression of carcinogenesis, epithelial–mesenchymal transition (EMT) has been repeatedly shown to be important in changing cell profile, metastasis, and even tumor recurrence [24,25,26]. Two genes, SLUG (a transcriptional repressor and regulator) and SOX10 (a TF), have been shown to have increased and preferential expression within TNBC. It has been shown that, when ALDH1 is simultaneously expressed with SLUG, there is an association with shorter disease-free survival rates and the conversion from EMT into mesenchymal–epithelial transition (MET) in metastatic sites. MET is the reverse of EMT, and it is denoted by the reacquisition of epithelial characteristics [27,28]. Of the TNBC subtypes, M and MSL have been linked to metaplastic BCs expressing some of these characteristics [29].

Another factor that contributes to the difficulty in treating TNBCs is the high degree of variance among the mutations acquired. The most common mutations are to p53 (up to 80%), followed by phosphatase and tensin homolog (PTEN) (35%) as well as inositol-polyphosphate-4-phosphatase type 2 (INPP4B, 30%) [30,31,32]. So far, none of the mutations described in TNBCs have been determined to be the “driver” of carcinogenesis [33]. The BRCA1 and BRCA2 genes have been shown to contribute to carcinogenesis particularly in BCs with a familial linkage (15–20%) or in those who are already carriers of one mutation in breast tissue [34,35,36]. Other risk factors are numerous, and lifestyle habits, such as diet and obesity, can increase the likelihood of developing TNBC [37]. During metastasis, TNBC has been documented to go to the lungs, liver, and brain more often than bone, which is the typical site for other BCs [38]. In a separate study, it was observed that, of 116 patients diagnosed with TNBC between 2000 and 2006 at the Dana–Farber Cancer Institute, 46% of the patients were diagnosed with metastases to parts of the central nervous system (CNS), and the median survival rate following prognosis was 4.6 months [39].

### 1.4. TNBC Ecology

Neoplastic cells must compete for resources with nearby healthy cells; however, these neoplastic clones also often compete amongst themselves. This is because different mutations acquired in these neoplasms impact replication and resource acquisition rates. It has been shown that neoplastic TNBC cell clones injected into opposite flanks of mice and rats can inhibit the growth of one another or that one set of clones can completely inhibit the other [40,41]. Neoplastic cells have also been described as parasites working towards establishing independence [14]. It has been suggested that the stimulation of fibroblasts and macrophages towards angiogenesis and the release of growth factors could aid TNBC clones in establishing tumors. The release of elastases and matrix metalloproteinases (MMPs) by macrophages for ECM degradation has also been suggested to be a contributing factor in establishing tumor beds and metastasis [42,43]. Additionally, it has been suggested that neoplastic TNBC clones participate in a commensal interaction with nearby non-metastatic clones where diffusible factors are absorbed, allowing them to take up a metastatic phenotype without needing to accumulate genetic mutations first [40,44,45,46].

Furthermore, data acquired from studying cocultures of different BC cell lines in vitro show that certain interactions between cells can impact growth, both positively and negatively. MDA-MB-231 cells coincubated with supernatants of MCF-7 cells increased proliferation over time, whereas coincubation with BT-474 cells or simply other MDA-MB-231 cells hampered growth [47]. Potential explanations for these observations were hypothesized as distant cell interactions where cell culture medium differences created competitively favorable environmental conditions or that ER^−^/PR^−^ cells can utilize auto and paracrine self-regulation pathways to synthesize their own GFs more frequently than BC cells that express ER and/or PR [48]. Therefore, these cells could independently sustain their growth without an autocrine loop. Neoplasms also have important interactions with the TME in supporting the growth of a developing tumor. These interactions are complex as differing TME conditions have varied impacts on carcinogenesis; normal epithelial cells can become carcinomas [49] and teratocarcinomas can have their phenotype suppressed [50] or even potentially reverted [51] based on conditions present in the TME.

### 1.5. Current Standard Therapies for TNBC

Due to the extreme heterogeneity of TNBC, chemotherapy with anthracyclines or taxanes remains the standard for the treatment of TNBC. There are a few mechanistic targets for therapy such as DNA repair, cell proliferation, and p53 [52]. Patients with TNBC have been documented to receive significant benefit from and even respond better to chemotherapy than other BC types do, despite the more aggressive phenotypes [53,54,55]. Ultimately, if a patient’s TNBC becomes metastatic, particularly during a recurrence, survival falls to 0%; even combinatorial therapies that trade increased toxicity for potentially elevated responses result in no benefit to survival rates [56,57].

However, in March 2019, the US FDA granted accelerated approval for an immunotherapy regimen for patients with BC. The treatment includes the use of atezolizumab (Tecentriq^®^) combined with nanoparticle albumin-bound paclitaxel (Abraxane^®^, nab-paclitaxel) in patients with metastatic TNBC where ≥1% of the tumor area is positive for immune cells expressing programmed cell death-ligand 1 (PD-L1) [58,59]. Atezolizumab is a humanized IgG1 monoclonal antibody that binds directly to PD-L1 and blocks a receptor’s interaction with PD-1 as well as the costimulatory protein B7-1 [60]. Nab-paclitaxel is a microtubule inhibitor that, when bound to nanoparticle albumin, increases solubility, minimizes hypersensitivity reactions, has better transport across endothelial tissue, greater tissue penetration, and a slower elimination time as compared to unbound paclitaxel [61,62,63,64,65,66]. Results from a double-blind, randomized, and placebo phase III clinical trial lead to this regimen’s accelerated approval [60]. When the combination of atezolizumab and nab-paclitaxel was compared with the placebo plus nab-paclitaxel for the primary treatment group, patients had an average increased progression-free survival (PFS) of 1.7 months and a 20% reduction in risk for disease progression and/or death. In the PD-L1-positive subgroup, the PFS increased further to 2.5 months as well as further decreased the risk for disease progression and/or death to 38% [60]. Even though this new therapy shows promise to increase the lifespan of patients slightly, this may merely delay the inevitable in most cases; therefore, new targeted therapies are still needed for TNBC.

## 2. Oncolytic and Immuno-Oncolytic Viruses (OVs)

These are a class of viruses that have either been selected as unmodified wild-type (*wt*) viruses or genetically engineered via the deletion of viral genes and/or the addition of immuno-modulatory genes for use as therapeutics against cancer. These viruses enter a tumor and ultimately lyse the cancer cells in a targeted manner, aiming to leave non-cancerous cells alone. Some OVs bear genetic inserts that induce changes within a cell, to the TME, or to immune cells within a host. Many of these viruses may also be modified via deletions to accomplish things such as decreased pathogenicity or increased tumor preference/selectivity [67]. As of now, there are 157 studies across the world that are using OVs in the treatment of various cancers in human trials, with 10 having TNBC as the host disease [68]; however, there are many more virus platforms “in the pipelines” for BC/TNBC. Some of these viruses are as follows: adenovirus [69,70], vaccinia virus [71,72], tanapoxvirus [73], parapoxvirus [74], marabavirus [75], poliovirus [76], measles virus [77], herpes simplex virus [78], and Newcastle disease virus [79].

### 2.1. History of Viruses in Cancer Therapy

Even though the term “oncolytic virus” did not immediately appear in the field of oncology, the first clinician credited for observing a virus inducing spontaneous tumor regression dates back to De Pace in 1912 [80]; he administered a rabies vaccine after a female patient was bitten by a dog, and her gynecological tumor began shrinking. There were also case reports from before De Pace, where a tumor would begin regressing alongside the incidence of a natural infection. In these reports, many of the patients had malignancies such as lymphomas or leukemia, both of which create significant immunosuppressive environments [81,82,83]. George Dock’s case has been widely referenced because there was a woman with leukemia who went into remission after being infected with influenza; her enlarged spleen and liver reduced in size almost back to normal, and she had a 70-fold reduction in leukocytes. The original report was made in 1896 and published in 1904, before De Pace, and nearly 40 years prior to influenza being categorized as a viral infection in 1933 [84].

Throughout the 1920s, Levaditi and Nicolau from the Pasteur Institute experimented on tumors transplanted into animals, mostly rabbits. They showed evidence that multiple “naturally occurring” viruses could induce the remission of implanted tumors, including complete regressions [85]. Newcastle disease virus (NDV) emerged as a virus of interest, originally isolated from poultry, by Levaditi and Haber, who more extensively described its oncolytic activity in 1936 [86]. In the late 1940s–50s, clinical trials commenced in the United States with naturally occurring viruses, such as dengue virus, West Nile virus, mumps virus, and NDV. These trials were abandoned, as most patients either suffered from viral disease in addition to their cancer or oncolytic effects were short-lived upon the development of antiviral antibodies [87].

Separately in 1949, there were two patients with Hodgkin’s lymphoma that experienced brief incidents of remission after becoming infected with hepatitis B virus. This generated a clinical trial where 22 patients with Hodgkin’s lymphomas were treated with impure samples of sera or tissue from individuals infected with “the hepatitis” virus [88]. Of the 22 patients, the first died directly from the virus and 13 others developed disease from hepatitis B; however, 7 of those who showed signs of viral disease had noticeable improvement in their cancer for at least 30 days, and 4 in that group even had tumor regressions. However, the trials with hepatitis viruses were stopped due to unacceptable toxicities.

One set of human trials with the mumps virus in Japan garnered much interest in the field. It was observed that host antiviral immunity enhanced the oncolytic activity seen in patients with ovarian cancer and glioblastomas [89,90]. Although the initial human trials had numerous issues, such as the selection of patients with pre-existing antiviral immunity and inconsistent sourcing for viral treatments, the results reported were promising; there were 90 patients treated, and 37 of them showed significant and even total tumor regression. Furthermore, 90% of the patients had some degree of response to the therapy, despite the presence of antiviral immunity in the majority of the enrolled patients [89]. Unfortunately, subsequent trials did not show similar levels of success, leading to the eventual halting of the pursuit of its use as a cancer therapeutic, until recently [90,91,92].

The expansion of chemotherapeutics, radiotherapy, and surgery, alongside a lack of funding, caused reluctance to continue clinical research on OVs in the U.S.A. With the U.S. president Nixon’s declared “war on cancer” in 1971, funding in the field was expanded for research on various cancer therapeutics, though for OVs funding was tighter than for other experimental therapies [85]. For progress to be made in the field, research had to be focused on developing better in vitro and in vivo animal models in order to provide evidence that oncolytic virotherapy was viable and, most importantly, safe in humans.

Attempts to address this issue had been developing since the mid-1940s by establishing animal models in order to test OVs. Initial reports showed that the newly available cell line mouse sarcoma 180 could be lysed by OVs in animals [93]. Russian Far East encephalitis virus (EEV) (known now as tick-borne encephalitis virus) [94] was used to treat transplantable mouse tumor cells, and observed tumors were completely eliminated with elevated doses in some cases. In 1951, another study described complete responses in experiments with five other mouse tumor lines [95]; it was observed that, after the tumors were infected and removed from one animal, the transplant would fail in another animal. A significant downside to EEV was that its neurotropism caused death for the treated mice. Regardless, these experiments set the precedent for using animal models in testing as a proof of concept prior to clinical trials in human subjects [96].

These experiments helped generate the hypothesis that viruses of non-human origin could be used in an oncolytic capacity without causing disease in non-adapted hosts. This would circumvent the obstacle of pre-existing antiviral immunity and pathogenicity [96]. Support for this idea came from work showing that the reverse was true; human viruses could be used to treat animal cancers [93,95], and work performed with NDV in Europe supported the idea that viruses from non-human natural hosts could be used as OVs in humans [86,97,98]. To increase efficiency and adapt viruses to human cells, repeated selective passaging was carried out in cancer cells. Data from EEV showed that it could increase oncolytic efficacy when passaged 20–30× in a tumor cell line as compared to the non-passaged parental strain [99]. It was also hypothesized that passaging in purified cancer cells would decrease tropism for healthy cells or avoid it altogether [100,101]. Another study was conducted where 24 candidate viruses were tested for replicative ability in human tumor cell lines; 6 viruses were identified with the capacity to do so in vitro [100]. Of those six, infectious bovine rhinotracheitis virus and equine rhinopneumonitis virus showed oncolytic activity in vivo in Syrian hamster models against HeLa cells and KB cells, respectively, further confirming the potential for viruses not endemic to humans to be candidates for development as OVs [101].

Other animal viruses were also explored, such as the avian plague virus (AVP), which was tested due to the belief that it would be less pathogenic to humans than many of the viruses used in the past were. A common side effect was strong neurotropism that would inevitably kill the host, even after destroying transplanted tumor cells [102]. The hypothesis that AVP was a safer and equally effective OV was tested in mice bearing sarcoma-97 tumors; the tumors were initially regressed and even caused some remissions, but AVP quickly spread to the brain of these mice and ultimately killed them [96]. Unfortunately, these data supported numerous other observations that every OV tested so far had some degree of neurotropism [99,103,104,105]. As described by Cassel and Garrett in a 1965 study, “When one considers the viruses that might be studied to derive a type which is oncolytic, non-neurotropic and has no major drawback, it is apparent that the list of candidate viruses is impressively small. A practical compromise would be to settle for an oncolytic virus with the lowest degree of neurotropism obtainable” [106]. NDV was selected as a potential OV for this study, where it was inoculated into mice with Ehrlich ascites carcinomas (EAC). Within 8 days, 97% of tumor cells were lysed. Considering that this was carried out with a very small inoculation of NDV (1 × 10^4^ EID_50_), and that there were no observable neurotropic effects, the results provided strong evidence that viruses were capable of cancer treatment. Further work still needed to be performed with potential OVs in order for them to become viable anticancer agents, as it was clear that many viruses were not suitable for therapeutic use in their natural forms.

### 2.2. Genome Manipulation and Genetically Engineered Viruses

Pathogenicity and neurotropism concerns with most OVs led the field to focus on decreasing tropism for non-cancerous tissues. An early review addressed the future of OV development [107]. One question was whether OVs could be manipulated in the lab to increase their oncolytic efficacy. The problem was that no technology was available for identifying genes of interest in the viruses, let alone modify them. Instead, it was suggested that tropisms for other tissues could be interfered with by other viruses or chemicals. It was well-known that viruses could interfere with the replication of one another [108], so experiments were carried out to see if NDV could reduce the neurotropism of viruses that showed great oncolytic potential, but later killed the mice due to encephalitis; most importantly, doing so while still retaining oncolytic efficacy at the tumor [109]. The investigation showed that the protection provided by viral interference was related to how quickly encephalitis took place. If encephalitis occurred rapidly, the protection was minimal from NDV; however, if the neurotropic process was slower, the protection was greater, but not complete. For Egypt 101 (West Nile virus), not only did NDV fail to prevent death from encephalitis, it also became less effective against the EAC tumors that it was supposed to treat. Alternatively, Bunyamwera virus encephalitis was prevented by NDV, but it failed to have therapeutic efficacy and the tumors killed the mice.

It took almost 30 years to see genetic engineering used for OVs, even though the principle was being used for gene editing therapies in other diseases [110,111]. The proof of concept for genetic engineering in viruses had been accomplished in the 1960s via the insertion of nucleotides to express polylysine in the tobacco mosaic virus [112]. It took until 1991 for genetic engineering to be used for OVs, with the deletion of viral thymidine kinase (TK) in herpes simplex virus-1 (HSV) for the treatment of glioma [113]. It was observed that TK-negative mutants of HSV were replicatively attenuated in normal, non-dividing cells, but could still replicate in cells that divided regularly, such as in cancer. Mice bearing U87 human gliomas were injected directly into their brains with the mutant HSV (dlsptk). Not only did the virus prolong survival, but it was capable of complete tumor elimination. This virus, although attenuated for neurovirulence, still had evidence of encephalitis in some mice. Despite this, more than 25% of the mice survived the cancer and OV injection [113].

Since then, the field of oncolytic virology has rapidly expanded, with the advent of genome sequencing to specifically target and remove pathogenicity genes [114], add in genes with immunomodulatory characteristics [73], and even modify viral surface proteins to allow the specific replication of an OV into tumor cells with corresponding receptors [77]. Genetic manipulation allowed for the attenuation of pathogenicity without crippling oncolytic efficacy, potentially even enhancing it. The threat of non-human OVs undergoing evolutionary pressure to switch host tropism exists, but modern screening techniques allow for the selection of viruses with lower mutational rates and higher genome stability.

## 3. Preclinical OVs for TNBC

Much research is being carried out in order to study several different OV platforms for the potential treatment of TNBC. OVs being studied in BCs not classified as TNBC will not be described here, though recent reviews of these viruses are available [115,116]. Below, currently described viruses tested against TNBC preclinically in vitro and in vivo, as well as those currently in human clinical trials, are summarized.

### 3.1. Adenovirus

Adenoviruses (ADV) are non-enveloped dsDNA viruses in the genus *Mastadenovirus* and the family *Adenoviridae*. At this moment in time, the largest number of OV recombinants from any viral classification that have been tested against TNBC are adenovirus strains. A type 5 adenovirus expressing green fluorescent protein (GFP) and the human telomerase reverse transcriptase gene (hTERT), OBP-401, was tested for infectivity in specialized MDA-MB-231 cells that had shown higher metastatic potential than their parental lines [117,118]. In vitro testing displayed increased GFP expression in the highly metastatic cells, which translated into lower percent cell viability in the highly metastatic cells than the parental cells [117]. It was also demonstrated that, following three treatments of OBP-401 in vivo, xenografted cells could be eliminated completely with no lymph node metastasis and very few lung metastatic cells in comparison to the vast metastasis seen in the control animals.

Another adenovirus, P55-HTERT-HRE-TRAIL, expresses TNF-related apoptosis-inducing ligand (TRAIL) with expression specificity to tumor cells driven by hTERT and hypoxia response element (HRE) promoters [69]. In vitro, the virus infected MDA-MB-231 cells preferentially over control MCF-10A cells in a significant manner, while also expressing TRAIL at elevated levels. When tested in vivo, the TRAIL expressing ADV inhibited orthotopic tumor growth, with increased efficacy at a higher dose (4 × 10^8^) and significantly prolonged survival in a metastatic tumor model, while also reducing metastases compared to control animals.

Combinatorial therapies and new approaches to the modulation of gene expression within infected tumors are also being tested in ADV platforms against TNBC. An oncolytic ADV was tested with an adjuvant therapeutic temozolomide, which is converted into an alkylating agent under physiological conditions. The combination was shown to enhance viral replication and oncolytic effect in TNBC cells in vitro [119]. Long non-coding RNAs have also been inserted into an ADV recombinant to disrupt metastatic cell signaling and oncogenic miRNAs that can help cancerous cells undergo EMT [120]. The resulting recombinant AdSVP-lncRNAi9 was effective at reducing tumor growth in mice and blocking the activity of multiple miRNAs that drive oncogenesis. Furthermore, a recombinant ADV expressing granulocyte-macrophage colony-stimulating factor (GMCSF) in combination with immune checkpoint inhibitors (ICIs) was able to control tumor growth in vivo and promote increased survival compared to control-treated animals [121]. Pairing histone deacetylase inhibitors (HDACis) with a recombinant ADV expressing an αvβ6-integrin-binding peptide from foot and mouth disease virus demonstrated effective tumor reduction and increased viral replication in a SUM159 xenograft mouse model [122].

Finally, a chimeric ADV has also been constructed to be tested as an oncolytic against TNBC. A chimeric virus between type 5 ADV and type 11b ADV was produced and the cytokine Rantes (regulated upon activation, normal T cells expressed and presumably secreted, also known as CCL5) was inserted into the E3 region of the viral genome [123]. The chimeric and recombinant Ad5F11bSP-Rantes was tested against a panel of TNBC cell lines in vitro and then trialed in an NCG mouse xenograft model bearing MDA-MB-231 tumors. The results from this study showed that the chimeric ADV efficiently reduced tumor size and that the expression levels of Rantes were elevated in the blood of mice treated with Ad5F11bSP-Rantes compared to a control and an eGFP expressing ADV. Added to this were significantly elevated levels of CD3^+^ T cells infiltrating tumors treated with the chimeric ADV, demonstrating that Rantes expression was helping to direct the chemotaxis of immune cells towards the tumor.

### 3.2. Herpes Simplex Virus

Herpes simplex virus (HSV) is a dsDNA virus from the genus *Simplexvirus*, family *Alphaherpesvirinae*. Canerpaturev (C-REV) is an oncolytic mutant of HSV that alone has antitumor efficacy against TNBC. It has also shown a promising oncolytic effect when combined with 5-fluorouracil (5-FU) and S-1 (a combination of gimeracil, oteracil, and tegafur) against TNBC in vivo in a bilateral 4T1 tumor model [124]. S-1 metabolizes into 5-FU and includes protective effects to increase the bioavailability of 5-FU within treated tissues [125]. The antitumor effect of C-REV is augmented by 5-FU via the latter enhancing the inhibition of growth factors, increasing the number of CD8^+^ T cells and their interferons, and decreasing the number of myeloid-derived suppressor cells (MDSCs). Reduced MDSC numbers are important as these cells suppress T cell activation, which would directly inhibit the antitumor effects driven by the immune response [126,127,128,129,130]. The study suggested that MDSCs should therefore be a cellular target when treating TNBC.

Talimogene laherparepvec (T-vec, Imlygic) is another HSV-1 mutant that has GMCSF inserted and deletes viral ICP34.5 as well as ICP47. The deletion of ICP34.5 is to decrease pathogenicity and make the virus tumor-selective [131]. ICP47 is deleted to prevent T-vec from replicating within cytotoxic T cells [132]. The deletion increases the expression of US11 within the virus, which increases its replication in tumor cells without compromising tumor selectivity [114]. Its mode of action is two-fold: the direct lysis of cancer cells, leading to the release of tumor-specific antigens (TSAs), and then the attraction of dendritic cells as well as macrophages to the TME from the release of GMCSF. The addition of the ICI paclitaxel as an adjuvant therapy stabilizes tubulin within microtubules and prevents depolymerization. This leads to mitotic arrest during the metaphase as spindle fibers cannot function properly [133]. When used together, they have been demonstrated to be a potent antitumor combination. Not coincidentally, T-vec is in clinical trials for many cancers and has been approved as the only oncolytic virus in the United States for the treatment of advanced-stage melanomas [114,134]. Though T-vec is currently in human trials for TNBC, no preclinical data on this cancer are available.

### 3.3. Chimeric Poxvirus

The original study detailing a chimeric Orthopoxvirus (dsDNA virus), CF33, describes the genome of this OV derived from nine different parental poxviruses: cowpox virus (Brighton strain), raccoonpox virus (Herman strain), rabbitpox virus (Utrecht strain), and six different vaccinia virus (VV) strains (WR, IHD, Elstree, CL, AS, and Lederle-Chorioallantoic) [135]. These viruses were coinfected into CV-1 cells, plaque-purified, and the isolates were then high-throughput screened for the highest tumorigenic properties against the cancer cell lines present in the NCI-60 lineup; CF33 was chosen from this screening. In vitro testing demonstrated potent cytotoxicity and viral replication in TNBC cell lines where endogenous phospho-Akt activity was present (BT-549, Hs578T, and MDA-MB-468). This activity results from mutations to the phosphatidylinositol 3-kinase (PI3K)/Akt pathway. The cell lines that demonstrated this activity allowed for better CF33 replication than in the MDA-MB-231 cell line, which has wtPI3K/Akt and PTEN pathways. When CF33 was tested in vivo in an MDA-MB-468 xenograft model, low titers of virus (10^3^ and 10^4^) demonstrated a significant reduction in tumor volume when compared to mock treated control tumors.

These results prompted another preclinical study where CF33 was modified to CF33-hNIS-anti-PD-L1 [136]. CF33-hNIS-anti-PD-L1 expressed two genes: sodium iodide symporter (hNIS) and a single-chain variable fragment of the PD-L1 binding ligand. CF33-hNIS-anti-PD-L1 also showed efficacy against MDA-MB-468 cells when tested in a xenograft model and did not show any dose-related toxicity. This included an intracranial injection at ten times the lethal dose of the Western Reserve (WR) strain. There was a lower degree of efficacy in syngeneic TNBC models 4T1 and E0771, but previous work had shown that the murine tumor cells were less susceptible to infection by CF33 [137]. Of note, the anti-PD-L1 antibody encoded by the virus was shown to be nearly as effective at competitive binding as the FDA-approved atezolizumab, with expression and functionality confirmed.

Another chimeric poxvirus has been developed and studied against TNBC. This chimeric virus, CF189, was created using a coinfection of cow kidney epithelial cells with orf virus strain NZ2 and the TJS strain of pseudocowpox virus [74]. CF189 caused lytic infection in numerous TNBC cell lines in vitro. MDA-MB-468 xenografts on mice were significantly reduced compared to the negative control with doses as low as 1 × 10^3^ PFU of CF189. Contralateral tumor sites on these mice were also controlled, even when the virus was not injected directly into those tumors, indicating the virus can travel to distal sites and exert antitumor effects.

### 3.4. Tanapoxvirus

Tanapoxvirus is a dsDNA virus from the genus *Yatapoxvirus* and family *Poxviridae*. It is of similar genomic and particle size to VV (144 kbp and ~65 nm, respectively), and causes mild febrile illness that is limited to humans and monkeys [138,139,140]. Preclinically, two virus recombinants have been tested in vivo against an MDA-MB-231 xenograft model in athymic nude mice, demonstrating significant tumor growth reduction: TPV/Δ66R/mIL-2 and TPV/Δ66R/mCCL2 [73]. The first recombinant carries inserted mouse interleukin-2 (IL-2), which serves primarily as the T cell and natural killer (NK) cell growth factor but also helps in T cell maturation as well as the activation of antitumor macrophages [141,142]; the second recombinant carries inserted mouse chemoattractant protein 1 (CCL2/ MCP1), which is an activator of monocytes, mast cells, dendritic cells, NK cells, and T cells in the inflammatory response [143,144,145]. Both viruses carry deletions of viral thymidine kinase (TK, ORF 66R), as is the case in many other poxviruses, to provide preferential replication in tumor cells. When tested in an athymic nude mouse model, both viruses showed significant differences in tumor volume compared to control mice and demonstrated increased immune cell infiltration into tumor sites as well as a reduced tumor mitotic index within the tumors compared to the control-treated tumors [73].

### 3.5. Vaccinia Virus

VV is a dsDNA *Orthopoxvirus* in the family *Poxviridae*. It is a prominent poxvirus in many cancer studies due to its large genome, which can handle many genetic inserts simultaneously, the fact that it replicates quickly, and due to it having a known safety profile. GLV-1h153 is an oncolytic VV that expresses hNIS. The natural expression of hNIS exists in tissues outside of the thyroid, including breast tissues; therefore, its overexpression in some cancers serves as a potential target, including in TNBC [71,146]. This treatment was paired with single-photon emission computed tomography (SPECT) imaging to track the uptake of radioactive iodine into tissues where the virus replicated. The radioactive iodine was also used as an adjuvant therapy for GLV-1h153, increasing its efficacy when used in combination in MDA-MB-231-xenografted mice compared to either treatment alone [71]. A similar virus, GLV-1h164, which encodes for GLAF-2 (an antibody against vascular endothelial growth factor, VEGF), was also tested in TNBC. Similar to the results with GLV-1h153, GLV-1h164 also showed efficacy in vivo in terms of treating MDA-MB-231 xenografts compared to a negative control and the parent virus (GLV-1h100). This virus also decreased blood flow to the infected tumors compared to parent-virus-infected tumors, demonstrating the effective targeting of VEGF [72]. DNA-based oncolytic viruses and genetic modifications employed therein, which have been tested preclinically in TNBC, are summarized in Table 1.

### 3.6. Alphavirus M1

M1 is a strain of the Getah-like alphavirus that was derived initially from mosquitos and is non-pathogenic in normal mice or rat tissues. It also has no known disease-causing capability in humans. It is a positive-sense RNA virus from the family *Togaviridae*. Initial studies showed that this virus had the ability to selectively kill cells lacking the antiviral zinc finger protein (ZAP) in human cancers. Oncolytic activity was reported to be mediated through stress to the endoplasmic reticulum leading to apoptosis [147]. Later, this virus was tested against varying types of BC, including TNBC [148]. In this study, M1 did not initiate anticancer activity as previously shown, but instead activated necroptosis. Though some pathways are shared between necroptosis and apoptosis, the inhibition of caspases and activation via tumor necrosis factor is required for necroptosis to occur [149]. This study also revealed that doxorubicin (a topoisomerase II inhibitor) activates the GAS6/AXL pathway of M1, enhancing replication and increasing oncolytic efficacy [148].

### 3.7. Coxsackievirus B3

Coxsackieviruses are positive-sense RNA viruses from the genus *Enterovirus* in the family *Picornaviridae*. A coxsackievirus B3 (CVB3) platform was used to generate a recombinant virus that expressed multiple targeting sequences (TSs) of microRNAs (miRNAs) to reduce off-target toxicities when used in vivo. The initial TSs selected were mouse miR-145 and miR-143 as they are significantly downregulated in tumor cells to assist in proliferation and cell division. Multiple copies of each were inserted into the 5′UTR of the CVB3 genome. Initial testing showed safety concerns by day 12 following intraperitoneal (IP) injections of recombinant miR-CVB3 [150]. Further modifications were then made to add additional tissue-specific TSs of miRNAs for muscle-specific miR-1 and pancreas-selective miR-216; the former is highly expressed in the heart of mice and the latter is highly expressed in the pancreas. Both have been reported to be downregulated in numerous cancer types, including breast cancers [151]. The new recombinant, miR-CVB3-1.1, demonstrated significantly better safety results and significantly less viral capsid protein detected in the target organs of the heart and pancreas than wtCVB3. When tested in a syngeneic murine TNBC tumor model bearing 4T1 tumors, miR-CVB3-1.1 showed a significant reduction in tumor volume when injected IT at a titer of 1 × 10^6^ when compared to an untreated control. The recombinant virus also significantly increased survival after 28 days compared to both untreated and wtCVB3 groups; however, intravenous (IV) administration failed to yield similar results to IT administration [150].

### 3.8. Maraba Virus

The oncolytic Maraba virus, MG1, is a genetically modified rhabdovirus (negative-sense RNA) originating from the vesicular stomatitis virus serogroup in the genus *Vesiculovirus*. Mutations were made to amino acid L123W in the M protein and Q242R in the G protein for attenuation in normal cells, while improving replication in cancer cells. This virus was also altered to include GFP [152]. After showing considerable efficacy in a mouse colorectal cancer model (CT26), with 100 percent survival and complete regression by day 35, the virus was tested in different BC lines, including patient-derived xenograft (PDX) models of TNBC [75]. MG1 showed variable infectivity in vitro of various BC lines but was greatly enhanced with paclitaxel treatment prior to infection. When the treatment was applied to a syngeneic mouse model bearing TNBC tumors, MG1 combined with paclitaxel significantly improved survival and reduced tumor volume in all three cell lines tested (EMT6, 4T1, and EO771) compared to a control, and in two cases when compared to MG1 alone [75]. Furthermore, when investigating metastases developed from TNBC in the same mouse models as before, MG1 was able to significantly reduce the number of metastases to the lungs. A combination with an ICI (anti-PD-L1 antibody) had the most promising results; the results showed that the combinatorial treatment had the smallest tumor loads and highest percent survivals when compared with any of their previous syngeneic TNBC mouse model experiments [153].

### 3.9. Measles Virus

Measles virus (MV) is a negative-sense RNA virus from the genus *Morbillivirus* in the family *Paramyxoviridae*. One oncolytic measles virus recombinant has been modified to include the proapoptotic gene BNiP3 in humans, rMV-BNiP3, which can act on BCL-2 family proteins and E1B 19 kDa proteins from adenoviruses [154]. This virus demonstrated a significant reduction in cell viability against MCF-7 and MDA-MB-231 cells in vitro when combined with paclitaxel and separately with a synthetic hydrazone derivative, H2, which is similar in structure to tamoxifen (estrogen modulator). Both H2 and paclitaxel, when combined with the oncolytic MV, significantly enhanced caspase 3 activity and apoptosis induction in MDA-MB-231 cells.

Another set of measles viruses was generated in order to target the urokinase cell surface receptor (uPAR), which is overexpressed in both mouse and human cancers [77]. One virus targets the human uPAR protein (MV-h-uPA) and the other targets the mouse homologous protein (MV-m-uPA). This occurs via the insertion of the corresponding amino terminal fragments of either human or mouse uPAR into the C-terminus of the mutated viral H glycoprotein [155]. Both virus recombinants were tested in NOD/SCID mice bearing MDA-MB-231 xenografts. The mice were injected three times, every other measurement period, and both viruses demonstrated the ability to individually regress the xenografts significantly when compared to a control. Furthermore, when both viruses were combined together in one treatment, an even greater synergistic effect was observed as the combinatorial treatment significantly reduced tumor volumes when compared to MV-h-uPA alone. The authors mention how MV-m-uPA cannot bind to the human uPAR present on the xenografted tumors but can bind to the mouse stromal cells and other mouse immune cells [156,157] that may have been recruited to the tumor [77].

### 3.10. Mumps Virus

Mumps virus is an enveloped, negative-sense RNA virus from the genus *Orthorubulavirus* in the family *Paramyxoviridae*. Samples of mumps virus used in the clinical trials described earlier in this review were obtained with the goal of continuing its development as an OV [158]. Since it was known that the original source material was not composed of purified viruses, the sample obtained from Japan was designated as MuV-U-Japan, and the subtypes within were isolated and studied for their oncolytic potential. Two of these isolates, named MuV-UA and MuV-UC, demonstrated potent oncolytic activity in vitro against stock TNBC cell lines and in vivo against an MDA-MB-231 xenograft model. The MuV-UC virus showed great potential in both tumor control and increasing survival when administered both IT and IV. In fact, multiple isolates alone, combinations of MuV-UA and MuV-UC, and even the MuV-U-Japan stock viruses showed significantly enhanced survival when delivered IV at 2 × 10^7^ median tissue culture infectious doses (TCID_50_), both as single-dose and multi-dose administrations. Conclusions drawn from this study support the idea of using MuV-UC for translation into clinical studies in the future.

### 3.11. Newcastle Disease Virus

NDV is a negative-sense RNA strain of *Avian orthoavulavirus 1* from the *Avulavirinae* subfamily and the *Paramyoviridae* family. A lentogenic strain of NDV, called LaSota, was used to determine the oncolytic nature and efficacy of this virus in combination with doxorubicin in TNBC. The in vitro results showed that this NDV strain could replicate in and cause the cytolysis of 4T1 cancer cells in a dose-dependent manner [159]. When tested in a 4T1 syngeneic mouse model, the virus was administered at various hemagglutination unit (HAU) doses from 32 to 128, with one group featuring a combination with doxorubicin at 64 HAUs. This combinatorial experimental group demonstrated the greatest tumor reduction by eliminating treated tumors by day 10. The further evaluation of visceral organs showed no toxicity from viral administration and no reported long-term effects 1 year post-treatment.

### 3.12. Reovirus

Reoviruses are segmented dsRNA viruses in the *Orthoreovirus* genus, subfamily *Spinareovirinae*. A reassortant r2Reovirus was genetically engineered to include nine segments from the T1L strain (serotype 1 Lang) and one segment from the T3D strain (serotype 3 Dearing) of reoviruses [160]. The r2Reovirus has shown the capability to infect BC cell lines with variable efficacy, with most efforts concentrated on MDA-MB-231 cells. With the addition of multiple topoisomerase inhibitors, including doxorubicin, r2Reovirus infectivity increased and infected cell viability decreased compared to a control treatment [161]. The r2Reovirus demonstrated that IFN III mRNA and protein were produced in infected MDA-MB-231 cells as opposed to type I IFN, with a combinatorial treatment with topoisomerase inhibitors having no effect on viral replication. Caspase activity is also activated as a result of reovirus infection, but not in the typical manner; in some cases caspase 9 is activated, but not caspases 3 and 7, to induce programmed cell death [160]. In a different study, doxorubicin was conjugated to the r2Reovirus in order to enhance cytotoxicity and investigate in vivo applications of this combination against TNBC in a syngeneic 4T1 mouse model [162]. Both r2Reovirus alone and r2Reovirus conjugated with doxorubicin significantly reduced tumor growth when compared to a buffer-treated control or doxorubicin alone. This model also produces metastases to the lungs of tumor-bearing mice, and, though not statistically significant, treatment with r2Reovirus alone or when combined with doxorubicin reduced metastatic 4T1 cells in the lungs compared to an untreated control. All of the listed RNA OVs and their genetic modifications are outlined in Table 2.

## 4. OVs in the Clinical Stage for TNBC

Oncolytic viruses that have reached human clinical trials for patients with TNBC are described below. A number of different approaches are being tested, including viruses being used as a monotherapy or combined with adjuvant chemotherapies, radiotherapy, or immunotherapies. A recent review of all available human trials involving OVs as of 2020 summarized the results and discussed the conclusions drawn from these studies [163]. Many trial studies have chosen combinatorial approaches with other commercial anticancer agents to be tested against each treatment individually, as some therapeutic regiments have seen improved clinical responses when compared to their components as monotherapies in other non-TNBC trials [164,165].

### 4.1. Adenovirus

ADV/HSV-tk is being tested in a phase II trial, with the objective being to test the efficacy as well as safety of stereotactic body radiation therapy (SBRT) together with the administration of ADV/HSV-tk and valacyclovir prior to treatment with pembrolizumab (Keytruda^®^) (NCT03004183). ADV/HSV-tk + valacyclovir has shown an increased antitumor activity of NK cells, the stimulation of T-cells, and promoted lymphocyte infiltration into treated tumors in a previous human trial for patients with recurrent ovarian cancer [166]. The addition of valacyclovir is used to inhibit viral DNA replication. This, in addition to SBRT, induces the “abscopal effect”, where antitumor immune responses are enhanced from the radiation, allowing localized treatment to have an impact on metastatic tumor sites [167]. Pembrolizumab binds to the PD-1 receptor and allows for the activation of self-reactive Th17 clones. Some preliminary results are available, but have not yet been analyzed fully.

A different adenovirus, MEM-288, is being tested in a phase I dose escalation study as a monotherapy against numerous solid tumors, including TNBC. MEM-288 carries human interferon beta and recombinant, chimeric CD40 (MEM40). The overall goal of this study is to determine the maximum tolerated dose within a 3-week period and its safety in patients. The secondary objectives include the determination of the recommended dosage for a phase II trial (where MEM-288 will be combined with an ICI), efficacy, progression-free survival, response duration, and the presence as well as nature of the antitumor responses elicited. Dosages in the preliminary trial will reach as high as 1 × 10^11^ adenovirus particles (NCT05076760).

### 4.2. Herpes Simplex Virus (T-vec)

T-vec has entered a phase I clinical trial (NCT04185311) for patients with TNBC, where the goal of the trial is to study the OV together with iplilmumab (an anti-CTLA-4 antibody) and nivolumab (an anti PD-1 antibody) prior to an attempted surgical resection. T-vec is to be injected intratumorally (IT) on three separate days (1, 22, and 36) with four nivolumab treatments as well as two ipilimumab treatments over the course of the study. In a different phase II study with T-vec, it was administered IT and combined with paclitaxel, followed by doxorubicin, cyclophosphamide, and surgery in patients with stage 2–3 TNBC. This study has been completed, with an analysis having been carried out of 37 of the 40 enrolled patients’ biopsies and trial outcome results (NCT02779855) [168]. The primary end point of the trial was met as 45.9% of patients having a grade 0 residual cancer burden index (RCB) and the RCB0-1 (minimal residual disease) rate being 65%, totaling 24 of 37 analyzed patients. In all of the enrolled patients, 4 had distant recurrence after 30 months with 1 death because of the cancer, though none of these recurrence events were from patients in the RCB0-1 group. Six weeks post-therapy, there was a significant increase in effector T cell and memory T cell populations compared to pretreatment, with 76% of patients also having an increase in CD8^+^ tumor-infiltrating T cells. This increased CD8^+^ cell population was significantly higher in RCB0-1 patients than RCB2-3 patients. Responses for patients with >1% PD-L1^+^ cells were positively correlated with treatment outcome, though not significant. A key conclusion from this study was that immunotherapy agents should be administered as closely as possible to the adjuvant chemotherapeutic to help take advantage of the activation of immune cells rather than waiting for residual disease or planning for sequential administrations after the chemotherapy regimen.

Additionally, another HSV-1, called ONCR-177, is being tested in a phase I trial against a multitude of cancers, including TNBC (NCT04348916). ONCR-177 expresses five different transgenes: interleukin 12 (IL-12), C-C motif chemokine 4 (CCL4, previously known as macrophage inflammatory protein), FMS-like tyrosine kinase 3 ligand extracellular domain (FLT3LG ECD), and anti-PD-1 as well as anti-CTLA-4 antibodies. It also does not have γ34.5 deleted, as is the case in most other HSV recombinant OVs. Previous work used miRNA silencing to allow the functional copy of ICP34.5 to exist in the viral genome and be inactivated at discretion; however, in cases where the interferon response may still function properly within a cancer cell, ONCR-177 would not be attenuated in these cells [169]. The goal of the clinical trial is to determine MTD and evaluate preliminary efficacy when ONCR-177 is injected IT either alone or when combined with pembrolizumab. Early data from this trial have led to the determination that the phase II dosage will be 4 × 10^8^ plaque-forming units (PFU)/4 mL every 2 weeks, for up to 10 total doses. No dose-limiting toxicity was observed, and the side effects were deemed to be mild, grade 1–2 [170].

### 4.3. Reovirus (Pelareorep)

Pelareorep (Reolysin) is a naturally occurring Dearing strain of reovirus serotype 3, and is currently in two different clinical trials for humans. The first is an early phase I clinical trial that includes the use of three other drugs, in addition to the oncolytic, in patients with various types of BC. In addition to pelareoreop, atezolizumab will also be used for patients with TNBC, administered on day 3 of treatment. The goal of this study is to determine the safety profile and elicitation of antitumor responses by the OV in early stage BC (NCT04102618). The second is a phase II clinical trial combining pelareorep with an ICI monoclonal antibody (retifanlimab) in late-stage TNBC (including metastatic TNBC) and other BCs (NCT04445844). The goal is to test the efficacy and safety of the treatment regimen, as well as whether the PD-L1 expression levels pre-, during, and post-treatment are indicators of treatment outcomes. In both cases, the OV is delivered IV, and the virus will be administered a total of four times in the planned 28-day increments (repeated cyclically as long as quality of life remains acceptable and disease progression does not occur in the phase II trial).

### 4.4. Vaccinia Virus

A phase I/IIa multicenter trial using BT-001, an oncolytic vaccinia virus encoding recombinant human anti-CTLA-4 and GMCSF, is being tested both alone and in combination with pembrolizumab. The phase Ia trial is exploring the dose escalation for BT-001 via repeated IT injections in many different advanced solid tumors with or without metastases as a monotherapy. Phase Ib will have BT-001 administered IV in combination with pembrolizumab in the same types of tumors. Finally, in the phase IIa trial, BT-001 will be administered IT while pembrolizumab is used IV (NCT04725331).

### 4.5. Measles Virus

A measles virus expressing hNIS (MV-NIS) is currently in a phase I clinical trial treating a variety of cancer types from squamous cell head and neck cancer to metastatic TNBC (NCT01846091). The primary objective of the trial is to determine how well the virus is tolerated in these patients and to assess the safety profile of the virus when administered IT. The study also aims to look for efficacy in treating the target cancers as well as tracking the expression of NIS throughout cancerous tissues using SPECT imaging [171]. MV-NIS displays selective oncolytic activity via CD46. CD46 serves as a membrane regulator of complement activation that is overexpressed in many types of human cancers [172,173,174]. CD46 is bound to the virus using the hemagglutinin protein, and infection causes syncytia to form, leading to lysis [175,176]. CD46 also functions as a cofactor for the inactivation of C3b and C4b of the complement cascade via Factor-I, which can protect the cell from apoptosis [177,178]. The virus was tested and shown to be effective in human patients when delivered IV with titers as high as 1 × 10^11^ capable of being tolerated in the treatment of multiple myeloma as long as the viral uptake was relatively slow [179]. MV-NIS is also genomically stable, with a low risk of reversion from its attenuated form into the pathogenic form [180].

Another phase I trial using a different recombinant, MV-s-NAP, is recruiting patients with metastatic breast cancers (NCT04521764). The primary objectives of this trial are to determine the overall safety profile, MTD, and cumulative immune responses to the virus and tumor following IT injection of the virus. Virus administration is set to occur up to three additional times following the initial injection, provided that there are no signs of unacceptable toxicities or disease progression. This virus recombinant expresses neutrophil-activating protein (NAP) from *Helicobacter pylori* and was tested for safety in mice prior to clinical assessment [181]. MV-s-NAP showed, when administered IT or IV, that mice sensitive to measles infection tolerated the injections well, having no signs of negative effects on body weight, liver function, or circulating proinflammatory cytokines. Though viruses were detected in multiple organs via quantitative real-time reverse transcription PCR, mice did not succumb to disease and continued growing as normal throughout the study period. OVs and the current status of their clinical trials in humans are summarized in Table 3.

## 5. Conclusions

This review summarizes the cellular biology of TNBC and the unique challenges that it presents to treatment. A brief history of OVs and their current studies in the pursuit of treating TNBC were also described. The combination of engineered tumor selectivity or replication preference, safety, low toxicity, and efficacy in vivo against a cancer type that makes conventional treatment extremely difficult would deem OVs as logical choices for further and continued investigation. Due to the extreme heterogeneity of TNBCs, many standard therapies have fallen short of achieving similar treatment success as compared to other BCs; all types except TNBC have seen increased 5-year survival rates over the past 20 years. The use of OVs with or without combinatorial drugs or radiotherapy has been taken to the clinic as a potential future treatment modality for TNBC and many other cancer types. Combinatorial regimens, such as the use of HDACis to enhance oncolytic HSV entry and replication within tumor cells [182], show that new pairings of already-existing therapies with OVs can lead to promising outcomes that can be translated quickly into human trials. Immuno-oncolytic viruses and other immunotherapies reflect a shift from a chemical to biological approach in the treatment of cancers. With T-vec being approved already in the United States for melanoma, other OVs will have the chance to gain approvals as well. 

## Figures and Tables

**Table 1 cancers-15-02393-t001:** DNA oncolytic viruses preclinically tested in vitro and in vivo against TNBC.

Virus Platform	Viral Recombinant	Genetic Modifications	Adjuvant Therapies	References
*Adenovirus*	OBP-401	hTERT GFP	-	[117,118]
	P55-HTERT-HRE-TRAIL	hTERT HRE promoter TRAIL	-	[69]
	OAd-mCherry	ΔE1ACR2 (Δ24) ADV serotype 3 surface receptor binding protein mCherry	-	[119]
	AdSVP-lncRNAi9	Synthetic ORF of nine oncogenic miRNAs expressed in TNBCs inserted into the E3 region	-	[120]
	rAd.GM	hTERT hGM-CSF	Atezolizumab Ipilimumab	[121]
	Ad5-3Δ-A20T	ΔE1ACR2 ΔE1B19K Surface receptor targeting of αvβ6 integrin	HDACis: Scriptaid MS275 Romidepsin Trichostatin A	[122]
	Ad5F11bSP-Rantes	Fiber knob sequence of Ad11b Rantes	-	[123]
*Herpes simplex virus*	Canerpaturev	ΔUL43 ΔUL49.5 ΔUL55 ΔUL56 Overexpression of UL53 and UL54	5-FU S-1	[124]
*Chimeric poxvirus*	CF33-hNIS-anti-PD-L1	Chimera of nine parental poxvirus strains (cowpox virus, raccoonpox virus, rabbitpox virus, WR, IHD, Elstree, CL, AS, and Lederle-Chorioallantoic strains of VV) hNIS Anti-PD-L1 antibody	-	[135,136]
	CF189	Chimera of orf virus strain NZ2 and pseudocowpox virus strain TJS	-	[74]
*Tanapoxvirus*	TPV/Δ66R/mIL-2	Δ66R (viral TK) Mouse interleukin-2	-	[73]
	TPV/Δ66R/mCCL2	Δ66R Monocyte chemoattractant protein 1	-	[73]
*Vaccinia virus*	GLV-1h153	ΔJ2R (viral TK) ΔA56R (viral hemagglutinin) Renilla luciferase-Aequorea green fluorescent protein fusion Beta-galactosidase Beta-glucuronidase hNIS	Radioactive iodine and SPECT imaging	[71]
	GLV-1h164	ΔJ2R (viral TK) ΔA56R (viral hemagglutinin) Renilla luciferase-Aequorea GFPfusion Beta-galactosidase Beta-glucuronidase Anti-VEGF antibody	-	[72]

**Table 2 cancers-15-02393-t002:** RNA viruses preclinically tested in vitro and in vivo against TNBC.

Virus Platform	Viral Recombinant	Genetic Modifications	Adjuvant Therapies	References
*Alphavirus*	M1	GFP	Doxorubicin	[147,148]
*Coxsackievirus B3*	miR-CVB3-1.1	Targeting sequences of oncogenic miRNAs (miR-145, miR-143, muscle-specific miR-1, and miR-216)	-	[150,151]
*Marabavirus*	MG1-GFP	L123W in M protein Q242R in G protein GFP	Paclitaxel Anti-PD-L1 antibody Anti-CTLA-4 antibody	[75,152,153]
*Measles virus*	rMV-BNiP3	BNiP3	Paclitaxel H2 compound	[154]
	MV-m-uPA MV-h-uPA	Retargeting to mouse (m) or human (h) urokinase receptor (uPAR)	-	[77]
*Mumps virus*	MuV-UA/MuV-UC	None	-	[158]
*Newcastle disease virus*	NDV (LaSota strain)	None	Doxorubicin	[159]
*Reovirus*	r2Reovirus	Chimera of T1L (serotype 1) and T3D (serotype 3) segments Doxorubicin conjugation	Topoisomerase inhibitors	[160,161,162]

**Table 3 cancers-15-02393-t003:** Oncolytic viruses being tested clinically in humans against TNBCs.

Virus	Phase	Title	Treatment Composition (OVs bolded)	Trial Status	Clinicaltrial.gov identifier
Adenovirus (MEM-288)	I	Study of MEM-288 oncolytic virus in solid tumors, including non-small-cell lung cancer (NSCLC)	**MEM-288**	Recruiting	NCT05076760
Adenovirus (ADV/HSV-tk)	II	SBRT and oncolytic virus therapy before pembrolizumab for metastatic TNBC and NSCLC (STOMP)	**ADV/HSV-tk** Valacyclovir SBRT Pembrolizumab	Active, not recruiting	NCT03004183
Herpes simplex virus (ONCR-177)	I	Study of ONCR-177 alone and in combination with PD-1 blockade in adult subjects with advanced and/or refractory cutaneous, subcutaneous, or metastatic nodal solid tumors, or with liver metastases of solid tumors	**ONCR-177** Pembrolizumab	Active, not recruiting	NCT04348916
Herpes simplex virus (Imlygic)	I	Ipilimumab, nivolumab, and talimogene laherparepvec before surgery in treating participants with localized, triple-negative, or estrogen receptor-positive HER2-negative breast cancer	**Talimogene Laherparepvec** Ipilimumab Nivolumab	Active, not recruiting	NCT04185311
Herpes simplex virus (Imlygic)	I/II	Talimogene laherparepvec in combination with neoadjuvant chemotherapy in triple-negative breast cancer	**Talimogene Laherparepvec** Paclitaxel Doxorubicin Cyclophosphamide Surgery	Active, not recruiting	NCT02779855
Reovirus (Pelareorep)	II	INCMGA00012 and pelareorep for the treatment of metastatic triple-negative breast cancer, IRENE study	**Pelareorep** Retifanlimab	Recruiting	NCT04445844
Reovirus (Pelareorep)	I	A window-of-opportunity study on pelareorep in early breast cancer (AWARE-1)	**Pelareorep** Letrozole Atezolizumab Trastuzumab	Terminated (enrollment concluded and primary objectives were met)	NCT04102618
Vaccinia virus (BT-001)	I/IIa	A clinical trial assessing BT-001 alone and in combination with pembrolizumab in metastatic or advanced solid tumors	**BT-001** Pembrolizumab	Recruiting	NCT04725331
Measles virus (MV-NIS)	I	Viral therapy in treating patients with recurrent or metastatic squamous cell carcinoma of head and neck cancer or metastatic breast cancer	**MV-NIS**	Completed	NCT01846091
Measles virus (MV-s-NAP)	I	A vaccine (MV-s-NAP) for the treatment of patients with invasive metastatic breast cancer	**MV-s-NAP**	Recruiting	NCT04521764

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
