# Peer review of "Triple-Negative Breast Cancer: Basic Biology and Immuno-Oncolytic Viruses"

_cancers, 2023, doi:10.3390/cancers15082393_

Round 1
Reviewer 1 Report
Monaco et al presented a revieew manuscript titled, "Triple Negative Breast Cancer: Basic Biology and Immuno- 2 Oncolytic Viruses". This review comprehended the applications of various oncolytic-immune-triggered anticancer activity of specific viruses. The manuscript covered the applications of OVs in multiple studies involving in vitro, in vivo, and in clinical trials. The manuscript is weel organized and well presented. I suggest following minor corrections, before its acceptance.
1. In line77: it was mentioned 'aMSCs give'. correct the typo error.
2. In lines: 297-300: plagiarism detected. Rephrase those sentences, if they are not standard definitions.
Reviewer 2 Report
Triple Negative Breast Cancer: Basic Biology and Immuno-Oncolytic Viruses
In this review, Michael L. Monaco and colleagues have described the cellular biology of Triple Negative Breast Cancer (TNBC) and the unique challenges it presents to treatment. They further provide a brief history of Oncolyric Viruses (OVs), describe OVs used for treatment of TNBC in both the preclinical setting, as well as in various stages of clinical development, and summarize strategies to enhance therapeutic potential. Given the extreme heterogeneity of TNBC, standard therapies have fallen short of achieving treatment success. With advances in viral genetic engineering and genome manipulation, in addition to promising OV preclinical studies in TNBC, there is renewed interest in the field of using OVs for the treatment of TNBC as seen by Phase 1 and 2 clinical studies, hence this review is timely. Overall, the work is well organized and balanced and instances that require clarity are listed in my comments below. The different sections are comprehensive and exhaustive with literature referenced from 1904 through 2022. The use of earlier literature to provide a historical context is understandable; however, there are references that are inaccurate, missing, irrelevant that have been noted.
A brief assessment of each section, and a few suggestions to improve the overall scientific accuracy of the information as well as help facilitate readers through the development of the review are provided.
Section1: The introduction is well structured and organized to understand the types TNBC, the receptors, the theories behind how TNBC is established and maintained, the pathogenesis and ecology. In addition, the therapies currently used to treat TNBC are discussed. My only critique for this section is to have more recent citations while discussing the statistics of the disease. For example, line 43 references cited include [1,2]. Reference 1 is Breast cancer statistics of 2017, although relevant a more recent citation should be used. SEER database or ACS can be used for a more recent reference. https://www.cancer.org/cancer/breast-cancer/about/types-of-breast-cancer/triple-negative.html
Section2: This section has provided a very thorough and comprehensive view of the history of viruses in cancer therapy with references that go as far back as 1904; however, inaccurate or references to be cited are listed below. In addition, the evolution of genetic engineering of viruses and manipulation of viral genomes to increase its oncolytic efficacy in tumors are described.
2.1, Line 244, Reference [86] cited is inaccurate as this references NDV virus and not the mumps virus study described. Remove this reference.
2.2, Lines 338-345, needs an appropriate reference to be cited.
Section 3:
3.1, Line 388, Ref [121] This reference is inaccurate. The literature referenced is a combinatorial therapy using Oncolytic Herpes Simplex virus and HDACis for the treatment of malignant melingioma not ADV for TNBC. Please correct to appropriate reference for ADV in TNBC. The reference Rodríguez et. al., (2022). Viruses 14 (5), 1006. 10.3390/v14051006 should be cited.
3.2, Lines 410-411, “This is crucial according to theF study because MDSCs suppress ……” Unclear statement. Please rephrase for clarity.
3.2, Lines 413-415 describes clinical trials in non TNBC cancers which are irrelevant to this section on preclinical OVs in TNBC. Since this is confusing to the reader, I suggest removing it.
3.3, Lines 469- 470 Instead of “Preclinically, two virus recombinants have been tested in vivo against MDA-MB-231 cells with significant tumor growth reduction, TPV/Δ66R/mIL-2 and TPV/Δ66R/mCCL2.”which reads like it was done in cell culture, I suggest “Preclinically, two virus recombinants have been tested in vivo against MDA-MB-231 tumor xenografted mouse models with significant tumor growth reduction, TPV/Δ66R/mIL-2 and TPV/Δ66R/mCCL”. This description provides the audience with an easier understanding of the scientific study. Also, the reference [73] should be cited here.
3.4. Line 483 or the first mention of Vaccinia virus should have its abbreviation (VV)
3.4. Lines 486-488 “… which expresses hNIS, as it is naturally expressed in tissues…..including breast tissues. Therefore, its overexpression …….… in TNBC [71,147] ” is unclear. An alternate explanation -“..oncolytic VV expressing human sodium iodide symporter (hNIS), that is naturally expressed in tissues…… including breast tissues. Therefore, overexpression of hNIS serves as …….… in TNBC [71,147].
3.5 Line 513, Reference [150] needs to be cited at the end of the sentence i.e., “ … oncolytic efficacy [150]”
3.6 Line 533 Reference [152] should be cited at the end of the sentence i.e., “ … IT administration [152].”
Section 4: provides a detailed overview of OVs used in clinical trials (mainly phase I and II) for TNBC either alone as a monotherapy or a combinatorial therapy with drugs or radiotherapy.
In some cases, goals and objectives of the clinical trials are stated first and then the recombinant oncolytic virus is described in other cases the OV is described first and then the objectives. It helps the reader follow along and understand if the structure is consistent.
Preclinical studies leading up to clinical trials are mentioned for some trials, it may be helpful to briefly touch on that for all clinical trials.
In addition, a few sentences describing when a monotherapy versus a combinatorial is advantageous in the context of the different types of TNBC would be worth commenting.
4.1, lines 618-619, the reference [162] cited is inaccurate and irrelevant to the clinical trial as it describes “Thymidine kinase gene therapy with concomitant topotecan chemotherapy for recurrent ovarian cancer”. Please sight appropriate reference.
Tables:
The Tables summarize oncolytic DNA and RNA viruses used in preclinical studies as well as provide a comprehensive overview of clinical trials in TNBC.
Table 1, Page 11, under the Adenovirus platform Ad5-3delta-A20T the Ref [121] cited is inaccurate as it describes combination of Oncolytic Herpes Simplex virus and HDACis for the treatment of malignant melingioma not ADV for TNBC. Instead the reference Rodríguez et. al., (2022). Viruses 14 (5), 1006. 10.3390/v14051006 is relevant to TNBC and should be included. Please correct to appropriate reference for ADV in TNBC.
Table 3- which describes the OVs tested in clinical trials should be presented with OVs in the same order of OVs described under Section 4 of the text for consistency and ease of reading. Clinical trials using measles virus in the table should be listed last following Vaccinia virus clinical trials.
References:
Line 941, Ref 77- Check the reference. The doi is inaccurate for the title and so is the authors and reference information. Accurate doi 10.1158/1541-7786.MCR-17-0016, authors include Jing et. al., and REF Mol Cancer Res. 2017 Oct; 15(10): 1410–1420.
In summary, I recommend the review with few minor revisions mentioned which improves the overall scientific accuracy and facilitates the synthesis of information for the reader.
